# Lameness in Cattle—Etiopathogenesis, Prevention and Treatment

**DOI:** 10.3390/ani14121836

**Published:** 2024-06-20

**Authors:** Renata Urban-Chmiel, Pavol Mudroň, Beata Abramowicz, Łukasz Kurek, Rafał Stachura

**Affiliations:** 1Department of Veterinary Prevention and Avian Diseases, Faculty of Veterinary Medicine, University of Life Sciences in Lublin, 20-033 Lublin, Poland; renata.urban@up.lublin.pl; 2Clinic of Ruminants, University of Veterinary Medicine and Pharmacy in Košice, Komenského 73, 04181 Košice, Slovakia; pavol.mudron@uvlf.sk; 3Department and Clinic of Animal Internal Diseases, Faculty of Veterinary Medicine, University of Life Sciences in Lublin, 20-033 Lublin, Poland; lukasz.kurek@up.lublin.pl; 4Agromarina Sp Z o.o., Kulczyn-Kolonia 48, 22-235 Hańsk Pierwszy, Poland; agriconsul@wp.pl

**Keywords:** cattle, dairy cows, lameness, veterinary prevention

## Abstract

**Simple Summary:**

Lameness is one of the most commonly observed health problems in cattle. It is a main cause of health and economic losses in dairy cattle farming. These problems usually affect the hind legs, due to the greater load on this part of the body. Given the polyetiological nature of limb diseases causing lameness in cattle and the resulting challenges to effective treatment and prevention, the aim of the study was to present the scope of the problem of lameness in cattle around the world and possible means of preventing and treating it. This review also focuses on the etiopathogenesis of lameness, its clinical symptoms, and methods of early detection.

**Abstract:**

The aim of this review was to analyse the health problem of lameness in dairy cows by assessing the health and economic losses. This review also presents in detail the etiopathogenesis of lameness in dairy cattle and examples of its treatment and prevention. This work is based on a review of available publications. In selecting articles for the manuscript, the authors focused on issues observed in cattle herds during their clinical work. Lameness in dairy cattle is a serious health and economic problem around the world. Production losses result from reduced milk yield, reduced feed intake, reproductive disorders, treatment costs, and costs associated with early culling. A significant difficulty in the control and treatment of lameness is the multifactorial nature of the disease; causes may be individual or species-specific and may be associated with the environment, nutrition, or the presence of concomitant diseases. An important role is ascribed to infectious agents of both systemic and local infections, which can cause problems with movement in animals. It is also worth noting the long treatment process, which can last up to several months, thus significantly affecting yield and production. Given the high economic losses resulting from lameness in dairy cows, reaching even >40% (depending on the scale of production), there seems to be a need to implement extensive preventive measures to reduce the occurrence of limb infections in animals. The most important effective preventive measures to reduce the occurrence of limb diseases with symptoms of lameness are periodic hoof examinations and correction, nutritional control, and bathing with disinfectants. A clean and dry environment for cows should also be a priority.

## 1. Introduction

Lameness in cattle is a serious health problem worldwide, affecting animal welfare and significantly contributing to culling. Enormous economic losses in the dairy industry result from reduced milk production and reproductive performance, medical costs, and increased risk of culling, death, and the development of other diseases. In meat production, apart from reduced growth, economic losses also result from reduced market value and product quality, high mortality, and premature slaughter [1].

Abnormalities in the way cows move are called lameness, which is often included among the classical clinical symptoms of the perception of pain in animals. It is manifested as locomotive disorders of varying severity, including asymmetrical movement, disturbances of rhythm, a slowed gait, reduced weight-bearing on the hooves, and incorrect posture. Lameness encompasses all limb and hoof diseases, both infectious and non-infectious, which through pain symptoms significantly reduce welfare parameters in animals, leading to reduced milk yield, reproductive losses, and even culling [2,3].

Hoof diseases described as lameness are most common in the first 3–5 months after calving [4], although minor symptoms are visible in the second month of lactation in 20% of cows in the herd [5,6].

Given the global and economically significant problem of lameness in dairy cattle, the aim of this review was to analyse the problem of lameness in dairy cows by assessing the health and economic losses. This review also presents in detail the etiopathogenesis of lameness in dairy cattle and examples of its treatment and prevention, with particular emphasis on an early diagnosis. 

## 2. The Problem of Lameness in Cattle around the World—Economic Aspects

Due to production technology, lameness is currently one of the major health problems in dairy cattle herds all over the world. Numerous studies conducted by research centres around the world have shown that in nearly 4000 dairy cattle herds over the last 30 years, the average percentage of lameness with a degree of advancement above 3 (on a scale of 1–5) has ranged from 5.1% to 45%, reaching up to 88% within individual herds [7].

In the United Kingdom, for example, the incidence of lameness in dairy cattle in a production season is 29.5%, while the overall lameness rate due to any cause was 30.9%. In 66.1% of cases it was due to white-line disease, in 53.2% to ulcers, in 53.6% to digital dermatitis, and in 51.9% to nonspecific lesions [8].

In Brazil, the rate is about 78% during the rainy season and 44% in the dry season, so it is clearly dependent on the time of year. By far the most common factor in the occurrence of lameness (12.4%) is interdigital dermatitis, while other limb diseases, such as double sole, chronic laminitis, sole ulcer, and interdigital hyperplasia, play a somewhat smaller role (7.4% to 9.8%) [9].

The incidence of lameness in dairy herds is highly variable and ranges from less than 1% to more than 50%, depending on the farm. In the United States, lameness in dairy cattle herds is observed at levels of 13% to 55% [10,11]. In many cases, the percentage of lameness in dairy cattle herds in the course of a year can reach even 70% [12]. At a given moment, about one quarter of cows are believed to be affected by lameness (25% prevalence). There are 55 cases of lameness per 100 cows in the course of a year (55% incidence rate) [4]. Practically all dairy cow hooves show past or present damage during inspection at slaughter. However, about 80–90% of causes of lameness in cattle are located within the hooves or toes, most likely due to the more variable load on the toe. In the vast majority of cases (about 75–85%), lameness affects the pelvic limbs.

Economic losses resulting from treatment costs, reduced production, and premature culling make up a high percentage of the total maintenance costs in dairy herds. The costs of lameness are influenced by the prevalence, incidence, and duration of lameness.

In Great Britain, for example, annual costs associated with lameness in herds of >100 dairy cows can range from £1715 (€2013.60) to even £7500 (€8805.80) [13]. According to data from 2009 [14], costs arising from the occurrence of specific hoof diseases amount to £518.73 (€352.26) in the case of sole ulcers, about £300 (€352.26) in the case of white-line disease, and about £154 (€180.86) in the case of interdigital lameness. By far the lowest cost was estimated in the case of digital dermatitis, at £75.6 (€88.79). For dairy herds in Denmark, the annual cost of lameness per cow is estimated at €307.50, with significantly higher costs incurred in cases of digital dermatitis (DD) [15].

In the United States, annual costs resulting from lameness range from $120 to $330 (€110.84 to €304.79) per cow, and from $100,000 to $200,000 (€92,350 to €184,700) in herds of 1000 cows [16]. However, according to research by Davis-Unger et al. [17], the total costs resulting from lameness in dairy cattle depend on the development and degree of advancement of the disease, the presence of additional clinical signs, e.g., oedema, or other diseases, and the season of the year. For example, the average cost of lameness was highest in cows with symptoms of lameness in autumn, without joint swelling, and amounted to $22.8/cow (€21.01/cow) [17].

In EU countries, average costs are highly variable and depend on prevention programmes. In the Netherlands, for example, the annual cost associated with lameness is estimated at about €27 per cow, and thus up to €3000 in the case of a herd of 100 cows. In Hungary, the average cost of lameness in herds in 2005 was about €62/cow, which translated to more than €6000 for a herd of more than 100 cows [18]. In Spain, the average annual costs of lameness caused by dermatitis, sole ulcer, and white line disease range from $11 to $51 (€10.16 to €47.01) per cow. An average herd of more than 60 cows with confirmed lameness incurred losses of about $700–3000 (€646.45 to €2770.50) annually, depending on the underlying type of lesion [19]. A study conducted in the Netherlands by Verhoef [20] showed that the average total costs arising from lameness in dairy cattle herds amounts to €3400–4600 annually, while the average annual cost per cow ranges from €60 to €83. On organic farms in the European Union, the average costs per cow associated with lameness are about €43, placing second after costs arising from mastitis (€96) [21].

Economic losses arising from lameness encompass three main production parameters: losses in milk production at an average level of about 40%, costs resulting from fertility disorders at about 30%, and treatment costs, also estimated at about 30%. One of the causes of high economic losses due to lameness in dairy cattle lies in the fact that it usually begins a few weeks or even months before diagnosis and lasts several weeks, or even up to five months, after the completion of treatment [22]. The percentage breakdown of costs arising from lameness in selected countries is presented in Table 1. 

The economic impact of lameness in dairy cattle also includes losses due to premature culling of highly productive cows, a reduction in the length of the lactation period by about two weeks, the birth of low-quality calves, and above all, reproductive disorders [4].

Selected deviations from normal reproductive parameters in dairy cows resulting from lameness are presented in Table 2.

## 3. Factors Involved in Hoof Diseases in Cattle

The etiopathogenesis of lameness in dairy cattle is multifactorial. Predisposing factors vary depending on the farm, region, and country. Infectious and non-infectious factors associated with maintenance conditions and herd management are most commonly mentioned. These include inadequate building size, excessive stocking density, unsuitable walking surfaces, e.g., slippery or slatted floors, sharp turns at the entrance or exit of the cowshed, or an unsuitable resting surface [2,39]. Detailed information on predisposing factors for lameness in cattle is presented in Table 3.

Infectious agents include a variety of bacteria, including anaerobic bacteria of the genus *Fusobacterium* spp., facultative aerobes of the genus *Campylobacter* spp., and aerobic bacteria such as *Staphylococcus* spp., *Streptococcus* spp. and *Treponema* spp. An environment predisposing to infections is also created by typically environmental bacteria such as *E. coli*, which are very often accompanied by *Staphylococcus* strains or bacteria of the family *Pasteurellaceae* [43], enabling colonization and replication of the pathogens directly responsible for inducing a given disease (Table 4).

Our own observations show that paronychia is a fairly common problem in Poland. It is usually caused by *Fusobacterium necrophorum*, but the most severe cases and the most difficult to treat are induced by *Trueperella pyogenes*, a pathogen often associated with mastitis. *T. pyogenes* is responsible for the highest percentage of cows eliminated from further use due to paronychia. It is observed in the first months of lactation. Cattle with paronychia exhibit lameness, usually in one limb. The hoof is swollen above the coronary band, and gaps and furrows with a yellow-brown, foul-smelling exudate appear in the interdigital spaces. Poor environmental conditions contribute to the development of the disease, which if left untreated causes permanent damage to the limb [44]. 

Research has shown that the incidence of infectious diseases depends on the individual resistance of animals in the herd. An example is digital dermatitis (DD, Mortellaro’s disease). Some cows in the herd are infected multiple times, while others from the same herd and their offspring show no symptoms of the disease. Holstein-Friesian cows and their crossbreeds are more susceptible to infections than other breeds [52,53]. In the case of lameness in older cows, the involvement of DD is less common, possibly due to increased immunity [39]. Mortellaro’s disease is a multifactorial disease encountered all over the world, in tie-stall and free-stall barns. Animals with this disease exhibit pronounced lameness and spend most of their time in a recumbent position. Lesions occur in the interdigital space or on the rear surface of the heels, mainly on the pelvic limbs. Inflammation is initially superficial and then develops into ulcerative granulomatous inflammation reaching the skin between the hooves. Conditions conducive to the development of the disease include excessive stocking density in the barn, poor hoof hygiene, or a poorly balanced feed ration. Untreated, the disease can lead to permanent damage to the limb [52].

Examples of the most common limb diseases in dairy cows are presented in Figure 1.

Significant predisposing factors for lameness, especially in HF dairy cattle, include high milk yield at peak lactation and nutritional deficiencies in the post-partum period [54,55]. Bran et al. [2] observed an increased frequency of lameness in HF dairy cows with BCS ≤ 3.0. On the other hand, Newsome et al. [56] found that lameness and the associated reluctance to use the feed table may be linked to a significant and rapid loss of body weight and a reduction in BCS.

The onset of lameness in cows is also influenced by nutritional parameters. According to Blowey [57], a suitable selection of nutrients such as calcium, copper, zinc, B vitamins (niacin and vitamin PP), vitamins A and D, amino acids (cysteine and methionine), and fatty acids, primarily linoleic and arachidic acid, is responsible for proper hoof horn production and hoof keratinization in cattle. An imbalanced diet leads to disturbances in the formation of the horn of the hoof wall and increases susceptibility to infectious diseases. Feeding cows easily digestible and high-energy feedstuffs with limited roughage may be conducive to metabolic diseases, including ruminal acidosis. Low ruminal pH (below 5) increases lactic acid production, causing disturbances of fermentation. Endotoxins resulting from these changes increase the production and release of histamine, causing constriction of the blood vessels in the hooves and a deterioration of their condition. A pathological consequence of these changes is the development of laminitis with symptoms of lameness [57]. 

An equally important factor contributing to lameness is the age of the cow [39]. Dairy cows in their second lactation or later, irrespective of breed (Jersey, HF, or crossbreds) have been observed to have various hoof anomalies contributing to gait disorders [2]. A study by Wilson et al. [58] found that the percentage of cows with evident lameness in their first lactation was 12.14%, as compared to 20.4% in cows in their second or third lactation. Observations conducted in varied dairy cattle herds in Ireland showed that with each year of age of the cows, the probability of lameness increased by about 20%. Moreover, a positive result for predicted transmitting abilities (PTA) for lameness increased its likelihood by about 37.5% [39].

An increase in the incidence of lameness, mainly in adult cattle, may result from chronic degeneration of the joints, ligaments, or bones playing an important role in correct posture and maintenance of body balance. Cows with large, bulging udders are also susceptible to lameness, due in part to their altered gait and uneven use of their hooves [59].

Access to pasture is generally believed to reduce the risk of lameness in cows in comparison to animals kept in livestock buildings throughout the production cycle [60,61]. The few studies assessing lameness in herds of grazing cows have shown that some lesions, such as sole ulcers, are diagnosed less often. However, certain elements of grazing systems, such as heat stress and the condition of the paths or cattle run, can potentially increase the risk of lameness [54,62]. It is also worth noting environmental factors such as high temperature and humidity, which can negatively affect hoof integrity. They are conducive to the formation and development of ulcers, the replication of microbes, and increased disease severity. These predisposing factors for lameness in cattle may vary depending on the maintenance system used on the farm. For example, a pasture system requires the implementation of additional measures enabling early detection of gait disorders, such as classification of the degree of lameness by manual methods, with thermal imaging cameras [63], or by visualization, as well as assessment of other cow activities, e.g., automatic gait and behaviour measurements [64].

Table 5 presents examples of methods of early diagnosis of lameness in cows.

Predisposing factors for lameness in dairy cows also include concomitant diseases directly affecting the limbs, including white-line disease (separation of the white line), sole injury and damage, e.g., sole and toe ulcers, digital dermatitis and inflammation of the interdigital space, heel erosion, and laminitis, caused by damage to the vascular tissue of the foot [2,67]. Statistical analysis of the effect of determinants associated with herd management on the occurrence of lameness has shown that when lameness is present in the herd, the likelihood of lameness in the next production cycle increases by as much as 47% in comparison with herds in which lameness has not been considered a major problem [39]. Risk factors at the herd level also include the size of the herd, the size of the grazing area, the presence of stones and slats in the pathways where cows move, and the angle of the inclination and turn of the route used by cows after milking [39].

## 4. Breed Determinants—Angle of Inclination of the Limbs (Holstein-Friesians)

Evaluation of the genetic predispositions resulting from breed determinants show that by far the highest rate of lameness is observed in Holstein-Friesian cattle and their crosses, e.g., in comparison to Jersey cows, irrespective of the maintenance system (pasture or cowshed) [2].

A very interesting study analysing genetic determinants and the frequency of lameness in dairy cows was presented by an international group of scientists from the University of Guelph in Ontario, Canada; Cornell University in Ithaca, NY, USA; and the Veterinary Service in Fort Collins, CO, USA [67]. Linear and threshold models were used to estimate correlations between genetic predispositions determined by breed and the occurrence of clinical lameness in dairy cows. Parameters such as herd size, stage of lactation, and parity were used in the statistical models. The analyses confirmed a significantly increased frequency of lameness in cows in early lactation, especially in older cows in their third lactation or later. Analysis of the anatomical structure resulting from breed determinants confirmed significant negative correlations (−0.76 and −0.64 for the two models) between the hoof angle and the frequency of lameness: the smaller the angle of inclination of the foot, the higher the frequency of lameness [67].

The correlation analysis also confirmed a significant relationship for the occurrence of lameness in cows that showed a tendency to walk or stand on the hock joints with the toes directed outwards. The genetic correlation between the angle of the hind limbs and the occurrence of lameness was −0.68 for the linear model and −0.64 for the threshold model [67]. The genetic correlation between rump width and the frequency of lameness was 0.60 and was also statistically significant, as in the case of the position of the pelvic limbs.

## 5. General Symptoms of Lameness in Dairy Cattle

Limb diseases in cattle are most often manifested by lameness, to an extent dependent on the advancement of the disease. Animals may manifest pain symptoms in a variety of ways, e.g., with a slower gait, a reduction in stride angle or length, step overlap, or asymmetrical strides (inconsistent gait) [64]. In considering the occurrence of clinical symptoms manifested by dairy cattle, many authors [68,69] use a five-point scale (Table 6) for describing the severity of lameness.

Lameness in dairy cattle is also manifested by disturbed behavioural patterns associated with rest and reduced feed intake, as well as disturbances of social behaviour involving isolation from the herd and increased aggression in other members of the herd towards the cow with lameness. Cows with lameness symptoms, apart from reduced milk yield and fertility, are also at greater risk of death. 

A study by Thorup et al. [42] showed a significant decrease in the total time the cow spends on feed intake. It was 84 min shorter on average in cows whose degree of lameness was ≥3 than in cows without lameness (197 min). A relationship between the degree of lameness in cows and changes in individual production parameters was also demonstrated by Kofler et al. [4] (2013), who confirmed that dry matter intake was significantly lower, by 3% to 16%, in cows with lameness ≥3–5°, while milk yield ranged from 1.56–1.12% and was significantly lower than in healthy cows (1.69%). Pain and discomfort experienced by dairy cattle with lameness also cause changes in physiological parameters, including intensification of stress responses or pro-inflammatory processes, which may also contribute to the occurrence of other diseases, especially infectious diseases [70,71].

Examples of changes in selected physiological parameters in cows with symptoms of lameness are presented in Table 7.

## 6. Treatment of Hoof Diseases and Therapeutic Interventions for Hoof Diseases

To treat lameness, veterinarians trim the hoof horn and clean toe wounds. Hoof trimming mainly serves to correct overgrown hoof horn, even out the thickness of the sole, remove dead tissue, and take weight off the diseased toe. Corrective hoof trimming is often sufficient to treat the initial stages of ulcers or traumatic injuries to the sole. It is also essential in the treatment of damage to the hoof horn and to remove foreign bodies from the horn. Following the removal of necrotic and loose tissue or weakened horn, an oxygen microenvironment is formed, preventing the replication of anaerobic microbes and the formation of abscesses. If the procedure is carried out without damaging the healthy tissue of the dermis, the pain following the procedure is minor, and the animal recovers more quickly [75,76,77]. 

After cleaning and corrective trimming of the hoof, protective dressings are applied (Figure 2). In clinical practice, the dressing is usually a thick layer of cotton wool soaked in copper sulfate or formalin, which is then secured with a bandage. In addition, in local treatment of hoof horn damage, a dressing with an antibiotic is used, usually tetracycline or oxytetracycline in powder form or sulfonamide. This type of treatment is used for toe damage, digital dermatitis, or atypical digital dermatitis. The dressing performs its function and protects the wound against infection provided that the animal stands on a dry surface. In most breeding environments, however, after the dressing is applied the cow is led onto a surface on which organic material (slurry or faeces) is present. In these conditions, the dressing becomes contaminated, which can lead to irritation of the healing lesions, thus prolonging treatment [78,79]. Another complication arises when the dressing on the limb is too tight. This can impede circulation in the hoof and healing of the wound, because the wound must be supplied with nutrients to heal properly. Some veterinarians [80,81] believe that applying a dressing can be harmful. They have often observed that damaged hooves heal more slowly in animals with a dressing than without one. In some situations, however, such as hoof haemorrhage, a dressing is essential. This dressing should be changed after 24 h or if it becomes badly soiled. Blood is a substrate for the replication of microbes, which is unfavourable for convalescence [80,82,83]. 

In the case of local hoof damage, ointments with a triple antibiotic and silver sulfadiazine based on white petroleum jelly are used. These preparations, in addition to their broad bactericidal spectrum, are used to maintain moisture, which aids in wound healing. Biotin supplementation has also been observed to improve hoof horn quality and prevent the recurrence of hoof lesions [84]. At the early stage of abscess formation in the hoof, the use of a long-acting antibiotic can improve the patient’s condition or prevent infection (Figure 2). If the infection is not contained and begins to progress, it can spread to the hoof joints. The result is difficult to treat, causing prolonged and painful inflammation of these joints. This leads to intense pain, resulting in lameness, unwillingness to stand, reluctance to eat, and reproductive disorders (pregnancy toxaemia). In such cases, the veterinarian should drain the pus, which will reduce the pain. Depending on the circumstances, an X-ray can be taken to better reveal treatment options. The area should be bandaged (protected from moisture) after drainage to keep the hoof dry for at least 24, and preferably 72, hours [84,85].

A different type of treatment of lameness is used in the case of paronychia. It should be initiated as soon as possible to avoid advancement of the disease. In the acute treatment process, some veterinarians [86] use benzylpenicillin as a first-line drug for three days or long-acting benzylpenicillin in a single injection. Penicillin treatment is used in many cases, and according to field data it is effective. However, a withdrawal period for milk is recommended after antibiotic treatment, which entails economic losses for farmers. For this reason, antibiotics that do not require a withdrawal period are increasingly used in veterinary medicine, depending on the severity of the disease. This type of antibiotic treatment enables the use of milk from cows and reduces economic losses. For this reason, many dairy farmers are interested in this type of treatment, but veterinarians should explain to them that treatment time will be prolonged, and that the treatment may be less effective. The use of broad-spectrum drugs has currently become widespread among veterinarians, mainly next generation cephalosporins, as drugs without a withdrawal period for milk. Nevertheless, in the face of growing antibiotic resistance, the use of systemic treatment does not always seem to be justified [86]. The Swedish researchers cited observations that a local dressing with salicylic acid can be used in the early stages of paronychia. This treatment reduced the temperature of the sick cattle, improved the animals’ overall condition, and reduced lameness within five days of its use. According to the authors, this type of treatment can be an alternative to antibiotics for treating uncomplicated paronychia in dairy cows. 

Since lameness causes pain in cows, one of the necessary therapeutic measures is to alleviate pain symptoms. The use of nonsteroidal anti-inflammatory drugs (NSAID) has been shown to be an effective way to relieve pain beyond corrective clipping and claw blocking [85]. For example, administration of ketoprofen for three days, from 24 to 36 h after calving in dairy cows, reduced the risk of lameness. The use of flunixin meglumine or oral meloxicam also had a positive effect, reducing symptoms of lameness [87,88]. According to available research [84,89], NSAIDs contribute to increasing the effectiveness of therapy in hoof diseases with an inflammatory process. They can also serve as a preventive measure against lameness and mobility problems in cattle, as most cases are caused by inflammatory processes and circulatory disorders in the hooves. It can also be noted that early use of NSAIDs in treating inflammatory processes in the hoof (i.e., treating lameness) can help reduce the number of hoof disease cases in subsequent stages of the animals’ lives.

## 7. Prevention of Lameness in Dairy Cattle

In the fight against lameness in cattle, the main focus should be placed on improving the animals’ living conditions. This means providing them with a good-quality surface to lie on, adequate space to stand up and lie down, and clean, dry stalls and floors. In a free-stall barn, sand or sawdust, or alternatively anti-slip mats, should be used on the floor in places where the animals rest and chew their cud. In tie-stall barns, if it is not possible to lengthen the stalls or secure the manure alley, mats should be used to allow the cows to lie down and stand up without slipping. On farms with a very large number of cows suffering from lameness, the farmer should be aware of the need to remove excrement frequently in order to maintain a dry surface. In addition, fresh layers of substrate should be added in places where cows are present. 

It is important to maintain an appropriate air temperature on farms, as high temperatures accelerate the development of infectious agents and facilitate their transmission between cows, while heat stress reduces immunity in cows. For these reasons, the temperature in cowsheds should range from −7 to 18 °C, at relative humidity of 60% to 80%, while for lactating cows it should range from 4 to 16 °C, depending on the relative humidity [84,90]. To maintain this temperature, it is necessary to install effective ventilation indoors, to cool off the animals, and even to use sprinklers when temperatures outside the barn are high. In buildings housing a high percentage of cows with lameness, the owners should consider creating cattle runs or pastures with shaded areas. This maintenance system will make it easier to keep the hoofs clean and well-functioning and to create places to bathe the hooves each time the cows exit or enter the barn.

A lameness prevention programme should be adapted for a given herd of dairy cattle. Such a programme may include the development of individual treatment protocols for the herd. A veterinarian can help to determine the most common and most costly causes of lameness in the herd and to develop specific prevention and treatment protocols for them. It is also important to train employees to quickly recognize and report cases of lameness. A veterinarian can also help to train workers responsible for treating routine cases using treatment protocols. Effective hoof examination can make use of a variety of mechanisms, such as visual inspection of the cow’s movement, palpation of the hoof [44,91], automatic measurement of touch and release angles [92], and the use of thermal imaging cameras (thermograms) [93].

In veterinary prevention of lameness in cattle, it is crucial that cows should not stand for long periods. A comfortable bed should be provided so that cows can lie down and chew for at least 10 to 14 h a day. Stall dimensions should be adjusted to the size of each cow. It is also extremely important to adapt the floor system, e.g., with soft bedding or appropriately profiled mattresses. In the case of concrete floors, it is essential to use grooving or add rubber floors or mats in feed alleys [94,95].

Controlling the quality of cows’ diet also plays a very important role in preventing lameness. This is facilitated by a balanced feed that meets the nutritional requirements of the cow, adjusted to its age, health condition, and stage of lactation. A constant rumen pH must be maintained as well. Therefore, the size of feed particles in diets such as TMR should enable chewing of the food to produce saliva, which also prevents acidosis by buffering acids in the rumen [85].

Hoof diseases, such as sole ulcer and white-line disease, are best resolved by improving the condition of passageways, e.g., by installing rubber strips in them, and/or examining individual groups of cows in different stages of production. A clean and dry environment for hooves should be a priority [62]. The surface cows walk on should be adapted to ensure their comfort; if cows do not lie down to chew their cud for at least 12 h, then they are standing or walking, which places an additional load on the structures of the ligaments and hoof horns. It is important to maximize the comfort of cows, particularly in the transition period, 2–4 weeks before calving and 2–4 weeks after. Proper care of the herd, particularly control of the animals’ movements on the farm, plays an important role in preventing lameness in dairy cows. Cows that are led to the milking parlour three times a day are especially susceptible to injuries, which take place when cows are forced to walk quickly [62].

Proper nutritional control plays an important role in preventing lameness in dairy cows [96]. Balancing highly concentrated feed rations in order to achieve production goals while minimizing negative health effects (such as hoof inflammation) can be compared to walking a tightrope. 

Research by Lischer et al. [96] assessed the effect of biotin on the healing of uncomplicated sole ulcers. The authors administered 40 mg biotin/day for 50 days to some of the animals included in the study. They observed that the healing process in these cows was faster, and the hoof horn was of better quality, which could reduce susceptibility to recurrence of the disease. 

Footbaths, commonly used to treat lameness caused by infectious factors, also play an important role in its prevention [97]. Measures should be taken to maximize the effectiveness of foot baths; for example, it is important to completely immerse each limb in the solution and to prolong the time the hoof is in contact with the disinfectant and conditioner, e.g., by immersing it 3–4 times in the preparation. Footbaths are on average 3–4 m long, 1 m wide, and 15–25 cm deep. They should be located in places where cows are driven, such as the passageway connected to the milking parlour or the exit of the milking parlour [98]. 

Examples of selected chemical agents used as ingredients in formulations for footbaths are presented in Table 8.

It is particularly worth emphasizing periodic hoof examination and systematic corrective trimming, even twice a year, especially in mid-lactation. This minimizes or prevents the development of claw horn disruption lesions and can significantly reduce cases of lameness in dairy cattle, by even 25% [100]. Preventive measures should also include efforts to improve the living environment of cattle, which may include considering the use of ecological products in the form of bacteriophage preparations in combination with selected essential oils or plant extracts [6].

Genetic methods also play an important role in reducing the incidence of lameness in cattle. For example, selective breeding of animals focused on conformational features such as a larger foot angle, the position of the pelvic limbs, the width of the body, and the shape of the udder, can significantly reduce the incidence of chronic lameness in cows [85,101].

## 8. Conclusions

Limb diseases manifested by symptoms of lameness have an enormous impact on the development of other health disorders in dairy cattle, especially problems with reproduction or milk production and the birth of weak calves. Disturbances of homeostasis of basic physiological parameters, especially parameters of the stress response or pro-inflammatory indicators, reduce animals’ immunity and increase their predisposition to bacterial and viral infections.

Systematic preventive measures play an important role in reducing the incidence of limb diseases in cattle. These include periodic limb examination in conjunction with hoof trimming, footbaths, nutritional control at each stage of production, training of personnel, and a clean and dry housing environment. Effective prevention not only reduces lameness in dairy cows, but also prevents recurrences of disease in cows that have previously suffered from lameness.

## Figures and Tables

**Figure 1 animals-14-01836-f001:**
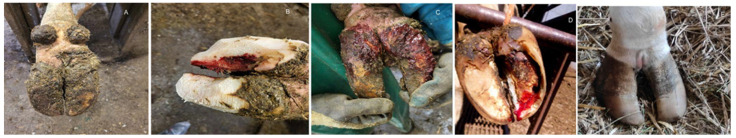
(**A**)—digital dermatitis, (**B**)—Rusterholz ulcer, (**C**)—inflammation of the hoof, (**D**)—white-line disease, (**E**)—interdigital hyperplasia.

**Figure 2 animals-14-01836-f002:**
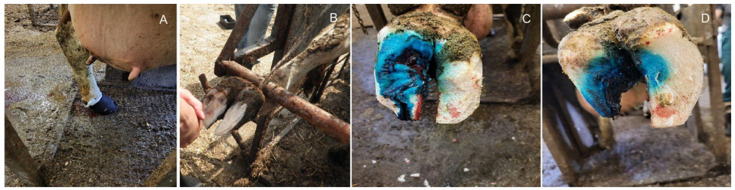
Examples of treatment of limb damage (sole ulcer) in dairy cattle. The infected corium and under-run horn were removed, sprayed with tetracycline (**B**–**D**), and wrapped in a bandage (**A**).

**Table 1 animals-14-01836-t001:** Costs resulting from lameness in various countries.

Country	Reduced Milk Yield (%)	Fertility Disorders (%)	Culling Costs (%)	Purchase of Drugs (%)	Veterinary Costs (%)	Other Costs	Total Costs (€)	References
United Kingdom	24	39	24	10	1	2		[14]
Netherlands	44	12	22	12	8	2	147–1393	[23]
Germany	26/44	nt	20	0.6	0.4	2	48	[21]
Spain	14	nt	16	0.2	0.1	0.6	31	
France	36	nt	14	0.6	0.3	2.4	53	
Sweden	13	nt	20	1	0.1	0.1	33	
USA	40	26–39	24	10	1	2	25.97–83.01	[24]

Legend: nt—not tested.

**Table 2 animals-14-01836-t002:** Documented impact of lameness on reproductive parameters in dairy cows.

Reproductive Parameters	Documented Effects of Lameness	References
Onset of signs of oestrus	8–9-day delay	[25]
	Interoestrus increased 2.8–2.9 × 92 vs. 82 days	[26]
	89 vs. 82 days	[27]
Calving-to-conception interval	On average 11–12 days longer	[28,29]
	113 vs. 93 days	[30]
	140 vs. 100 days	[31]
	180 vs. 130 days	[32]
	134 vs. 104 days	[33]
	163 vs. 119 days	[34]
First service	3–4 days longer	[35]
Calving interval	2% longer, significantly prolonged—absence of or weak oestrus	[36,37]
Anoestrus	Frequency increased 15.6×	[27]
Fertility rates	Lower conception rate, 41% vs. 55%	[31,33,38]
	Lower conception rate (0.52×)	
	Lower first-service conception rate 10% vs. 43%	
Services per conception	9× the average number	[26]
	On average 5 vs. 3	[31]
	2.45 vs. 2.15	[27]
	1.35-fold risk of unsuccessful insemination	[32]

**Table 3 animals-14-01836-t003:** Non-infectious predisposing factors for the development of lesions in the limbs causing lameness in dairy cattle.

Predisposing Factor		References
Environmental	Poor sanitation in breeding environments. Excessive stocking density (ease of transmission). Environment conducive to injury. Poor stall construction (too short or too narrow; excessive slope). Heat stress. Incorrect herd management. Concrete floors. Wet floors. Improper hoof correction. Lack of hoof inspection. Keeping animals in draughts. Concomitant limb and hoof diseases. Season of the year.	[40,41,42]
Individual	Size of cow. Hoof pigmentation. Age of cow. Herd size.	
Dietary	High-energy fodder. Insufficient fibre in diet. Excessive protein in diet. Mineral or vitamin deficiencies. Sudden changes in diet, especially in the post-partum period.	
Concomitant diseases	Metabolic disorders, e.g., subacute ruminal acidosis (SARA), leading to laminitis and white-line ulcers.	
Genetic	Inherited hoof structure. Breed predispositions, mainly in Holstein-Friesian cows.	

**Table 4 animals-14-01836-t004:** Infectious factors inducing hoof diseases with symptoms of lameness.

Type of Illness	Isolated Infectious Agents	References
Digital dermatitis	*Fusobacterium* spp., *Bacterioides* spp., *Campylobacter* spp., *Peptococcus* spp. *Treponema* spp.	[43,44]
Paronychia (chronic hoof infection)	*Fusobacterium necrophorum*, *Bacteroides nodosus*. Possible involvement of *Dichelobacter nodosus*, *S. aureus*, *E. coli*, *Trueperella pyogenes*	
Interdigital dermatitis	*Dichelobacter (Bacterioides) nodosus*, *F. necrophorum*, *Treponema* spp.	[45,46]
Interdigital phlegmon (foot rot, interdigital bone marrow necrosis)	*F. necrophorum*, *D. nodosus*	[46,47]
Heel horn erosion	*F*. *necrophorum*, *D. nodosus*	[43]
Arthritis	*Mycoplasma* spp.	[48]
Bovine respiratory disease complex	*Mycoplasma* spp., *Histophilus somni*, *Klebsiella pneumoniae*, *Clostridium* spp.	[49,50,51]
Skin and multi-organ infections	*Staphylococcus* spp., *Streptococcus* spp.	[43]

**Table 5 animals-14-01836-t005:** Examples of methods of diagnosis of lameness in dairy cattle and their effectiveness.

Diagnostic Method	Specificity %	Sensitivity %	References
Thermal imaging (depends on temperature)	28.1–77.8	55.4–92.9	[63]
Visual techniques	76–91	-	[64]
5-point scale 1–2° 3° 4–5°	60 90.9 81.2	-	[65]
Automatic gait and behaviour measurements			
Measurement of walking cows	75–96	-	[63]
Measurement of standing cows	>96	96.4	
Pressure-sensitive position mat Stride length, stride time, stance time, step overlap, abduction, left vs. right limb asymmetry, step width asymmetry, step length asymmetry, step time asymmetry, stance time asymmetry, relative step force asymmetry	86–100	85–90	[64]
Measuring gait and/or activity with accelerometers	75–96%	-	[66]

**Table 6 animals-14-01836-t006:** Scale of advancement of lameness in dairy cattle with a description of symptoms [68,69].

Assessment of Animal’s Movement	Effect on Animal’s Movement	Clinical Symptoms		
		Animal at Rest	Moving Animal	Head Position
No lameness, healthy animal	The cow moves freely, with its body weight distributed evenly on its four limbs.	When the cow is standing, the back is straight and its legs and hooves are correctly positioned.	Normal gait. The back remains straight.	The head is held stable in line with the back or slightly below it, both at rest and while walking.
2. Minor lameness	The cow’s movement is limited, but not greatly.	The back is straight when the cow is standing.	The back is slightly arched when the cow walks. The gait is disturbed. Body weight is not evenly distributed on the limbs and the cow may exhibit slight lameness while walking.	The head may be slightly lowered, briefly or continuously.
3. Moderate lameness	The cow moves with difficulty, taking short steps (with one or several limbs).	The cow stands with its back arched and its legs and hooves may be incorrectly positioned.	Gait is disturbed. The back is arched when the cow walks and it exhibits moderate lameness.	The head moves up and down during movement.
4. Lameness	The cow’s movement is limited. It spends more time recumbent.	The cow stands with its back arched. The limbs are incorrectly positioned.	The cow walks with a pronounced limp, with its back arched, favouring the affected limb; lameness is clearly evident	The head moves up and down during movement and is held low at rest.
5. Severe lameness	The cow’s movement is limited; it stays in a recumbent position and is reluctant to stand up.	The back is severely arched. The cow is unwilling to stand and to put weight on at least one limb; it may have difficulty getting up from a recumbent position or standing up at all.	The cow clearly favours the affected limb. The back is arched and the cow moves with difficulty and takes short steps. Severe lameness is evident, and the cow may vocalize.	The head moves significantly upwards and especially downwards when the cow walks. The head is lowered when the cow is at rest.

**Table 7 animals-14-01836-t007:** Selected physiological parameters in cows with symptoms of lameness.

Parameter	Healthy Cows	Cows with Lameness Symptoms	*p*-Value	References
Cortisol µg/L	28.6	63.5	*p* < 0.001 *	[72]
FCMs ng/g	40.4	54.5	0.3	
Norepinephrine pg/mL	680.31	967.3	*p* < 0.001 *	[70]
β-endorphins	42.92 pg/mL	67.75	*p* < 0.001 *	
DHEA ng/mL	1.52	2.35	*p* ≤ 0.001 *	[73]
Pulse	77.7	66.7	<0.001 *	[70]
Total protein conc. g/L	84.9	86.7–89.5	*p* = 0.05	[74]
Albumin	41.3	43.3–39.0		

* statistically significant differences; DHEA—dehydroepiandrosterone; FCMs—faecal cortisol metabolites.

**Table 8 animals-14-01836-t008:** Chemical agents used for hoof baths (Bednarski, [99], our own modification).

Substance	Concentration	Properties	Side Effects
Formalin	3–5%	Disinfecting and drying	Carcinogenic properties
Copper sulfate CuSO_4_	3–5%	Disinfecting and drying; hardens the hoof horn	Potential toxic effects in cows Negative environmental impact
Zinc sulfate ZnSO_4_	5% (10–20%)	Disinfecting and healing; strengthens the hoof horn	None
Other: copper nitrate, urea,	Depends on substance	Disinfecting and healing; strengthens the hoof horn	Copper nitrate is slightly irritating.

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
