# Peer review of "Lameness in Cattle—Etiopathogenesis, Prevention and Treatment"

_animals, 2024, doi:10.3390/ani14121836_

Round 1

Reviewer 1 Report

Comments and Suggestions for Authors

Lameness in cattle—etiopathogenesis, prevention and treatment 

 The review fills a gap in the literature and demonstrates appropriate construction. Keywords: Replace the highlighted items, they are not part of the title.

Effectively highlights the space related to the clinic of ruminants, addressing various aspects related to the hoof health of ruminants.

The methodology is in accordance with the presented results.

The conclusion should be rewritten, removing the citations and restructuring the entire content to align with the objectives of the study. 

The references are appropriate.

Figure 1: Add a scale bar to each of the images on the board. Also include the corresponding letters A-E. Figure 2: Add a scale bar to each of the images on the board. Also include the corresponding letters A-D. Table 6: Review formatting.

Author Response

Lameness in cattle—etiopathogenesis, prevention and treatment
The review fills a gap in the literature and demonstrates appropriate construction. Keywords: Replace the highlighted items, they are not part of the title.

Effectively highlights the space related to the clinic of ruminants, addressing various aspects related to the hoof health of ruminants.
The methodology is in accordance with the presented results.
The conclusion should be rewritten, removing the citations and restructuring the entire content to align with the objectives of the study.
The references are appropriate.
Reply: Thank You very much for Your opinion. According to the suggestions we have made a slight corrections of the manuscript. We have added also an additional text with the information suggested by other reviewers.

Figure 1: Add a scale bar to each of the images on the board. Also include the corresponding letters A-E. Figure 2: Add a scale bar to each of the images on the board. Also include the corresponding letters A-D.
Reply: The letters A-E and A-D on Fig. 1 and 2 has been added.
Table 6: Review formatting.
Reply: done

Reviewer 2 Report

Comments and Suggestions for Authors

A good and sound work . I added only very small changes that should be addressed.

In this article the etiopathogenesis, prevention and treatment of lameness in cattle was reviewed

The topic is very relevant in the field of preventive medicine in farm animals, because of animal welfare, but also in terms of the additional costs caused by orthopedic diseases, which can also lead to premature culling of the animals.

This sound review of literature is important in finding good and promising strategies to reduce lameness related problems in cattle.

In my opinion the study design and methodology is sufficient.

Conclusions are consistent with the evidence and arguments presented and they address the main question posed

The references are appropriate?

Tables and figures are ok.

8. Additional comments:

Line 192-211: The costs are given in £, € and $ which makes it difficult to compare – it would be nice to write the costs also in € for £ and $.

Line 326: it is written: These cites Swedisch researchers… - do you mean: these cited Swedish researchers?

Author Response

8. Additional comments:
Line 192-211: The costs are given in £, € and $ which makes it difficult to compare – it would be nice to write the costs also in € for £ and $.
Reply: Economic losses resulting from treatment costs, reduced production, and premature culling make up a high percentage of the total maintenance costs in dairy herds.  The costs of lameness are influenced by the prevalence, incidence and duration of lameness.
In Great Britain, for example, annual costs associated with lameness in herds of >100 dairy cows can range from £1715 (€2013,6) to even £7500(€8805,8) [39]. According to data from 2009 [40], costs arising from the occurrence of specific hoof diseases amount to £518.73 (€609,09) in the case of sole ulcers, about £300 (€352,26) in the case of white-line disease, and about £154 (€180,86) in the case of interdigital lameness. By far the lowest cost was estimated in the case of digital dermatitis, at £75.6 (€88,79).
In the United States, annual costs resulting from lameness range from $120 (€110,84)to $330 (€304,79) per cow, and from $100,000 to $200,000 (€92350 to €184700) in the case of a herd of 1000 cows [41].
In EU countries, the average costs are highly varied and depend on prevention programmes. In the Netherlands, for example, the annual cost associated with lameness is estimated at about €27 per cow, and thus up to €3000 in the case of a herd of 100 cows. In Hungary, the average cost of lameness in herds in 2005 was about €62/cow, which translated to more than €6000 for a herd of more than 100 cows [42]. In Spain, the average annual costs of lameness caused by dermatitis, sole ulcer and white line disease range from $11 to $51 (€10,16 to €47,01) per cow. An average herd of more than 60 cows with confirmed lameness incurred losses of about $700–3000 (€646,45 to €2770,50) annually, depending on the type of lesion causing it [43]. A study conducted in the Netherlands by Verhoef [44] showed that the average total costs arising from lameness in dairy cattle herds amount to €3400–4600 annually, while the average annual cost per cow ranges from €60 to €83.
On organic farms in the European Union, the average costs per cow associated with lameness are about €43, placing second after the costs arising from mastitis (€96) [45].
Economic losses arising from lameness encompass three main production parameters and involve losses in milk production at an average level of about 40%, costs resulting from fertility disorders at a level of about 30%, and treatment costs also estimated at about 30%. One of the causes of high economic losses due to lameness in dairy cattle lies in the fact that it usually begins a few weeks or even months before diagnosis and lasts several weeks or up to five months after the completion of treatment [46]. The percentage of costs arising from lameness in selected countries is presented in
Line 326: it is written: These cites Swedisch researchers… - do you mean: these cited Swedish researchers?
Reply: Yes it was corrected. Thank You very much for Your comment.

Reviewer 3 Report

Comments and Suggestions for Authors

The authors have attempted to develop a narrative review of the etiology, risk factors, treatment, and prevention of lameness in dairy cattle. This is an important topic that requires an extensive review of the literature to facilitate comprehensive findings and a good understanding of lameness management in dairy herds.

The fact that the authors attempted to look at studies published globally is really commendable. However, I have a few suggestions on how to improve the manuscript.

Introduction

The introduction is very superficial and fails to capture the contents of the review. The objective of the review is not well-stated in the section, and it remains unclear why the paper was developed and the contribution to the present knowledge pool on lameness in dairy cows.

Subsections of the review

The first section after the introduction is titled "Etiological factors of lameness" but the contents include both risk factors, etiopathogenesis and diagnostic methods (as shown in the last Table), which then raises the question of the mismatch with the heading.

Etiological factors for limbs and hoof lesions should include vital and updated information on the infectious and non-infectious causes of lameness. Subsections may be dedicated to discussing these 2 broad categories of causes of lameness and the specific hoof lesions - particularly DD and foot rot for infectious and claw horn lesions (white line disease, sole ulcer, and sole hemorrhage) for the non-infectious group.

- The prevalence and distribution of the causes of lameness in dairy cows can be provided on a global scale, with reports from studies conducted in tropical and sub-tropical countries.

- the section on the diagnosis of lameness should be moved to a more appropriate section

-        I think the section on “Problem of lameness in cattle around the world” should be moved to the early sections in the articles

-        Same suggestion applies to the section on the Economic impact of lameness.

-        The authors may consider moving these two sections to the first sub-topic after the introduction before moving to the etiological factors and specific causes of lameness

-        The literature sources are too few, there are several studies on the economic impact of lameness and specific claw lesions, that need to be incorporated to build a more robust and comprehensive review

-        Again, how do you transit from economic impacts of lameness to symptoms of lameness? Please the authors should re-arrange the flow of the sub-topics. I suggest symptoms should be discussed alongside the specific causes of lameness because the manifestation of lameness varies according to the specific claw lesion---

-        “Therapeutic Treatment of hoof diseases” - might consider changing to simply “treatment of hoof diseases or therapeutic interventions for hoof diseases”

-        Again, the content is very superficial and lacks comprehensiveness given the abundant literature that the authors could use in developing the section. There is also a discussion on the prevention of lameness in the section despite mentioning only “therapeutic” in the heading

-        The section on the prevention of lameness is also very superficial. The discussion should be tailored in such a way that the specific measures for infectious and non-infectious hoof lesions are clearly defined.

-        Finally, it will be worthwhile to provide brief information on the methods used in selecting the articles included in the review. I understand that this is a narrative review, but at least, the readers should be informed on the methods applied.

-        - There is no section to discuss the limitations of the review, which I consider very crucial to highlight the reasons why the information provided is extremely superficial. The manuscript has to be revamped before any significant contribution to lameness research can be attained. 

Comments on the Quality of English Language

Moderate revision required

Author Response

Introduction
The introduction is very superficial and fails to capture the contents of the review. The objective of the review is not well-stated in the section, and it remains unclear why the paper was developed and the contribution to the present knowledge pool on lameness in dairy cows.
Reply: Thank You very much for Your comments. We have added more detailed information about the lameness health problem and we have also created main aim of this review. In this part of the manuscript we also added the additional necessary references.

Subsections of the review
The first section after the introduction is titled "Etiological factors of lameness" but the contents include both risk factors, etiopathogenesis and diagnostic methods (as shown in the last Table), which then raises the question of the mismatch with the heading.
Etiological factors for limbs and hoof lesions should include vital and updated information on the infectious and non-infectious causes of lameness. Subsections may be dedicated to discussing these 2 broad categories of causes of lameness and the specific hoof lesions - particularly DD and foot rot for infectious and claw horn lesions (white line disease, sole ulcer, and sole hemorrhage) for the non-infectious group.
Reply: Thank You very much for Your suggestions we have made a proper correction of the title of this section according to Your comment. And now the title is more general for this section.
- The prevalence and distribution of the causes of lameness in dairy cows can be provided on a global scale, with reports from studies conducted in tropical and subtropical countries.
- the section on the diagnosis of lameness should be moved to a more appropriate section
-        I think the section on “Problem of lameness in cattle around the world” should be moved to the early sections in the articles
-        Same suggestion applies to the section on the Economic impact of lameness.
-        The authors may consider moving these two sections to the first subtopic after the introduction before moving to the etiological factors and specific causes of lameness
Reply: According to the suggestions we have transferred these sections in previous early section of introduction. Thank You very much for Your suggestions. We really appreciate Your help.
-        The literature sources are too few, there are several studies on the economic impact of lameness and specific claw lesions, that need to be incorporated to build a more robust and comprehensive review
Reply: We have added the proper literature to this section according to the suggestions. Thank You very much for Your comment, we really appreciate Your work.
-        Again, how do you transit from economic impacts of lameness to symptoms of lameness? Please the authors should rearrange the flow of the sub-topics. I suggest symptoms should be discussed alongside the specific causes of lameness because the manifestation of lameness varies according to the specific claw lesion---
Reply: We have made changes to the order of subsections and now general clinical symptoms are placed immediately after etiological factors, which allows for better understanding of the text. We also have made a slight correction of the subtitle of this chapter.
-        “Therapeutic Treatment of hoof diseases” - might consider changing to simply “treatment of hoof diseases or therapeutic interventions for hoof diseases”
Reply: This subtitle has been changed according to the suggestions. Thank You very much for Your comment.
-        Again, the content is very superficial and lacks comprehensiveness given the abundant literature that the authors could use in developing the section. There is also a discussion on the prevention of lameness in the section despite mentioning only “therapeutic” in the heading
Reply: Thank You very much for Your suggestions. We have made a proper correction of the title of this subchapter. We have also added proper references with the text in the text of this subchapter according to the comments. Thank You for Your help.
-        The section on the prevention of lameness is also very superficial. The discussion should be tailored in such a way that the specific measures for infectious and non-infectious hoof lesions are clearly defined.
Reply: In this section we also added the additional information about the prevention of lameness including nutrition, floor with proper references. Thank You very much once again for Your suggestions.

Round 2

Reviewer 3 Report

Comments and Suggestions for Authors

-              Why is the simple summary blank??

Abstract

-the authors failed to revise the abstract and most of the issues raised were left untouched. 

- What is the aim of this review?

- Methodology used in the review should be provided

- Findings should be well-spelt out

- Limitations of the review should be provided? 

Introduction

-              There is a moderate improvement in the introduction section but some of the comments raised earlier were not addressed. The authors should kindly provide a justification for not responding to most comments on the introduction, particularly in terms of flow.

-              The last sentence stated this review also partially presents the authors’ research findings. Is it necessary to include such information in the introduction? What are the authors trying to achieve by providing such an information? This is a review on lameness and its management, prevention, and treatment…I don’t see the need to pinpoint your article in this section, except if you are only looking at studies conducted in a particular country or region.

-              The statement On organic farms in the European Union, the average costs per cow associated with 113 lameness are about €43, placing second after the costs arising from mastitis (€96) [21]” is too short to be a paragraph. Consider merging with the previous or the next paragraph.  

-              The heading for Table 1: Costs resulting from lameness in selected countries. There is no need for selected countries, except you want to justify why those specific countries were selected. Is it that publications from other countries could not be retrieved or characterised by low quality?

-               Factors Etiological factors involved in limb and hoof diseases in cattle. Were the reviewed studies also explored limb diseases as causes of lameness or only hoof diseases?? If the latter is the case, then I suggest removing limb diseases from the heading  

-              What do the authors mean by “error in herd management” ??

-              Please be careful when discussing the role of nutrition in lameness onset. The statement “The onset of lameness in cows is unquestionably influenced by nutritional parameters” should be revised because this is not applicable to all types of hoof diseases in dairy cows. For instance, the onset of infectious hoof diseases like DD and HHE may occur without any nutritional influences. There are recent publications on the infectious agents that invade the digits even in cows without any nutritional imbalance. Moreover, non-infectious hoof lesions such as Sole ulcer and Whiteline disease can develop as a result of hormonal imbalance occurring during the calving period and weakening the locomotor apparatus in the hoof capsule. This can also occur even in well-fed cows with good BCS. That’s the reason I suggested that risk factors should be discussed under infectious and non-infectious hoof diseases, rather than merging them altogether. 

-               

-              Treatment of hoof diseases or therapeutic interventions for hoof diseases should be presented using one term (either treatment or therapeutic) – kindly remove the OR

In the concluding part under section 6, more information should be provided on the role of NSAID in the treatment of lameness in dairy cows. When it is indicated, and if it is required before or after preventive hoof trimming.  Kindly remove the full stop after “the use of.”

-              A limitation of the review is very important. 

Comments on the Quality of English Language

Moderate revision required

Author Response

Dear Editors,

I am attaching the revised article, which has also been proofread by a Native Speaker.

- our responses to the Reviewer's comments:

Comments and Suggestions for Authors

-              Why is the simple summary blank??

AD: We have add the Simple Summary according to the suggestion. Thank You very much for Your comment.

Abstract

-the authors failed to revise the abstract and most of the issues raised were left untouched.

- What is the aim of this review?

- Methodology used in the review should be provided

- Findings should be well-spelt out

- Limitations of the review should be provided?

AD: Thank you Very much for Your comments. The abstract has been rewritten accoring to the suggestions contained in all questions.

Introduction

-              There is a moderate improvement in the introduction section but some of the comments raised earlier were not addressed. The authors should kindly provide a justification for not responding to most comments on the introduction, particularly in terms of flow.

AD: Thank You very much for Yopur comments., we reallyu appreciate Your help. We have made a proper corrections according to the previous comments as well as to this comments. However the Introduction should be a short part of the manusript according to the other reviewers, so we decided to leave this part with only slight correction.

-              The last sentence stated this review also partially presents the authors’ research findings. Is it necessary to include such information in the introduction? What are the authors trying to achieve by providing such an information? This is a review on lameness and its management, prevention, and treatment…I don’t see the need to pinpoint your article in this section, except if you are only looking at studies conducted in a particular country or region.

 AD: This sentence has been changed according to the comment.

-              The statement “On organic farms in the European Union, the average costs per cow associated with 113 lameness are about €43, placing second after the costs arising from mastitis (€96) [21]” is too short to be a paragraph. Consider merging with the previous or the next paragraph. 

 AD: This part of the text has been changed accordin to the suggestions. Thank You very much for Your help.

-              The heading for Table 1: Costs resulting from lameness in selected countries. There is no need for selected countries, except you want to justify why those specific countries were selected. Is it that publications from other countries could not be retrieved or characterised by low quality?

AD:  It was changed

-               Factors Etiological factors involved in limb and hoof diseases in cattle. Were the reviewed studies also explored limb diseases as causes of lameness or only hoof diseases?? If the latter is the case, then I suggest removing limb diseases from the heading 

AD: It was changed

-              What do the authors mean by “error in herd management” ??

AD: It’s mean that the herd management procedures are inapropiate for example the bioasecuration, feeding or other. We have made a proper correction in this part of the text. Thank You very much for Your help.

-              Please be careful when discussing the role of nutrition in lameness onset. The statement “The onset of lameness in cows is unquestionably influenced by nutritional parameters” should be revised because this is not applicable to all types of hoof diseases in dairy cows. For instance, the onset of infectious hoof diseases like DD and HHE may occur without any nutritional influences. There are recent publications on the infectious agents that invade the digits even in cows without any nutritional imbalance. Moreover, non-infectious hoof lesions such as Sole ulcer and Whiteline disease can develop as a result of hormonal imbalance occurring during the calving period and weakening the locomotor apparatus in the hoof capsule. This can also occur even in well-fed cows with good BCS. That’s the reason I suggested that risk factors should be discussed under infectious and non-infectious hoof diseases, rather than merging them altogether.

AD: Thank You very much for Your comments. It was changed

-              Treatment of hoof diseases or therapeutic interventions for hoof diseases should be presented using one term (either treatment or therapeutic) – kindly remove the OR

AD: Treatment of hoof diseases or and therapeutic interventions for hoof diseases

In the concluding part under section 6, more information should be provided on the role of NSAID in the treatment of lameness in dairy cows. When it is indicated, and if it is required before or after preventive hoof trimming.  Kindly remove the full stop after “the use of.”

AD: According to available research (84, 89), NSAIDs contribute to increasing the effectiveness of therapy in hoof diseases with an inflammatory process. They can also serve as a preventive measure against lameness and mobility problems in cattle, as most cases are caused by inflammatory processes and circulatory disorders in the hooves. It can also be noted that early use of NSAIDs in treating inflammatory processes in the hoof (i.e., treating lameness) can help reduce the number of hoof disease cases in subsequent stages of the animals' lives.

-              A limitation of the review is very important.

AD: Yes we know about it, so we decided wrire only the most important details. Thank You very much for Your suggestions and all comments.

Kind regards,

Beata Abramowicz PhD, DVM
